# Diastereodivergent chiral aldehyde catalysis for asymmetric 1,6-conjugated addition and Mannich reactions

Wei Wen [1,3 ✉], Ming-Jing Luo[1,3], Yi Yuan [2], Jian-Hua Liu [1], Zhu-Lian Wu [1], Tian Cai [1], Zhao-Wei Wu [1], Qin Ouyang [2 ✉] & Qi-Xiang Guo [1 ✉]

Chiral aldehyde catalysis is a burgeoning strategy for the catalytic asymmetric α-functionalization of aminomethyl compounds. However, the reaction types are limited and to date include no examples of stereodivergent catalysis. In this work, we disclose two chiral aldehyde-catalysed diastereodivergent reactions: a 1,6-conjugate addition of amino acids to *para*-quinone methides and a bio-inspired Mannich reaction of pyridinylmethanamines and imines. Both the *syn*- and *anti*-products of these two reactions can be obtained in moderate to high yields, diastereo- and enantioselectivities. Four potential reaction models produced by DFT calculations are proposed to explain the observed stereoselective control. Our work shows that chiral aldehyde catalysis based on a reversible imine formation principle is applicable for the α-functionalization of both amino acids and aryl methylamines, and holds potential to promote a range of asymmetric transformations diastereoselectively.

[1] Key Laboratory of Applied Chemistry of Chongqing Municipality, and Chongqing Key Laboratory of Soft-Matter Material Chemistry and Function Manufacturing, School of Chemistry and Chemical Engineering, Southwest University, 400715 Chongqing, China. [2] College of Pharmacy, Third Military Medical University, 400038 Chongqing, China. [3] These authors contributed equally: Wei Wen, Ming-Jing Luo. ✉email: wenwei1989@swu.edu.cn; ouyangq@tmmu.edu.cn; qxguo@swu.edu.cn

Compounds containing multiple stereocentres can be found extensively in natural products, drugs and biology. Both the absolute and relative configuration of stereocentres can greatly affect biological activity[1]. Selective access to all stereoisomers is important to chemists, biologists and pharmacologists alike. Over recent decades, great strides in asymmetric synthesis have been made with biocatalysis, transition metal catalysis and organocatalysis. Those achievements have led to numerous methods for the enantioselective synthesis of optically active molecules[2]. Generally, the absolute configuration of the product can be controlled by choice of the enantiomer of the chiral catalyst[3,4]. The selective generation of individual diastereomers is often more challenging: typically, one diastereomer is inherently preferred, and other diastereomers cannot be efficiently produced via the same strategy[5]. Diastereodivergent catalysis is a wonderful concept which can overcome this diastereoselective bias in the chiral induction process. To date, several elegant catalytic asymmetric diastereodivergent strategies have been disclosed[6–8]. Diastereodivergence in asymmetric catalysis has been achieved by changing reaction conditions and reaction procedures, or by employing an entirely different catalytic system[9–14]. Many reports of diastereodivergence are serendipitous, but catalyst-induced diastereodivergence is more predictable and, as a result, more useful. Several elegant catalytic asymmetric diastereodivergent reactions controlled by either a single catalyst or two catalysts have been disclosed in recent years[15–19].

Chiral aldehyde catalysis is a burgeoning strategy for the creation of novel asymmetric reactions of amines[20–25], especially for the catalytic asymmetric α-functionalization of aminomethyl compounds[26–29]. However, only four reactions have been successfully realised using this strategy, all of which employed amino acids as the sole type of nucleophiles[30–33]. The chiral aldehyde-catalysed α-functionalization of aryl or alkyl-substituted aminomethyl compounds has not been reported, and no examples of diastereodivergence in chiral aldehyde catalysis have been disclosed. In 2014, we reported that chiral binaphthol (BINOL)-derived 3-formyl aldehyde I catalysed α-alkylation of 2-aminomalonates with 3-indolylmethanols[30]. In 2018, we reported another chiral aldehyde, 2-formyl BINOL aldehyde II, catalysed nucleophilic addition of glycine derivatives to α,β-unsaturated ketones[31] (Fig. 1). An investigation into the mechanism of these transformations indicated that both chiral aldehydes I and II use the formyl moiety to activate donors and the hydroxyl moiety to activate acceptors. Interestingly, the distribution of these two key moieties related to stereoselective control was conformationally distinct in chiral aldehydes I and II.

As shown in Fig. 1, by fixing the formyl plane and observing from the bottom, the Brønsted acid site (2′ hydroxyl) of chiral aldehyde I resides above the plane. Conversely, the steric hindrance group (R) is above in chiral aldehyde II. To navigate possible hydrogen bonding and steric effects, an acceptor would likely approach the formyl plane from different sides in chiral aldehydes I and II, and thereby expose a different face to react with the donor. Thus, we anticipated that I and II would be good candidates for the investigation of stereodivergence in chiral aldehyde catalysis.

In this work, we first test a diastereodivergent 1,6-conjugate addition of amino acids to para-quinone methides catalysed by chiral BINOL aldehydes I and II, respectively. Both anti- and syn-conjugate additions can be achieved with high yields, diastereo- and enantioselectivities. We then seek to expand the scope of this catalytic system by investigating a bio-inspired Mannich reaction of pyridinylmethanamines and imines. The success of these two transformations suggests that it may be possible to achieve diastereodivergent chiral aldehyde catalysis for a whole series of asymmetric organic reactions using catalytic systems derived from BINOL aldehydes I and II.

## Results

**1,6-Conjugated addition**. para-Quinone methides are versatile building blocks in asymmetric catalysis, particularly as Michael acceptors reacting with various nucleophiles[34–41]. The catalytic asymmetric nucleophilic α-addition of amino acids to para-quinone methides is an elegant strategy for the construction of optically active diarylmethanes and unnatural amino acid derivatives. However, current methods depend on the use of amino acid-derived Schiff bases or azlactones as nucleophiles[38–41]. In light of the unique properties of chiral aldehyde catalysis that it can employ N-unprotected amino acids as reactants and the potential for a diastereodivergent catalytic system using of 3- and 2-formyl BINOL aldehydes I and II, the 1,6-conjugate addition of amino acids to para-quinone methides was chosen for initial investigation.

We began by evaluating the reaction of para-quinone methide **1a** with tert-butyl glycine ester **2a** catalysed by chiral aldehyde **3a** and **4a**, respectively. Tetramethyl guanidine (TMG) was used as base, and toluene as solvent. As expected, when chiral aldehyde **3a** was used as catalyst, anti-**5a** was generated in 49% yield, 81:19 dr (diastereomer ratio) and 33% ee (enantiomeric excess) (Table 1, entry 1); when catalyst **4a** was employed, syn-**5a** was generated in 36% yield, 64:36 dr and 58% ee (Table 1, entry 2). Inspired by these preliminary results, we then optimised the reaction conditions of the anti- and syn-reactions independently. Reaction

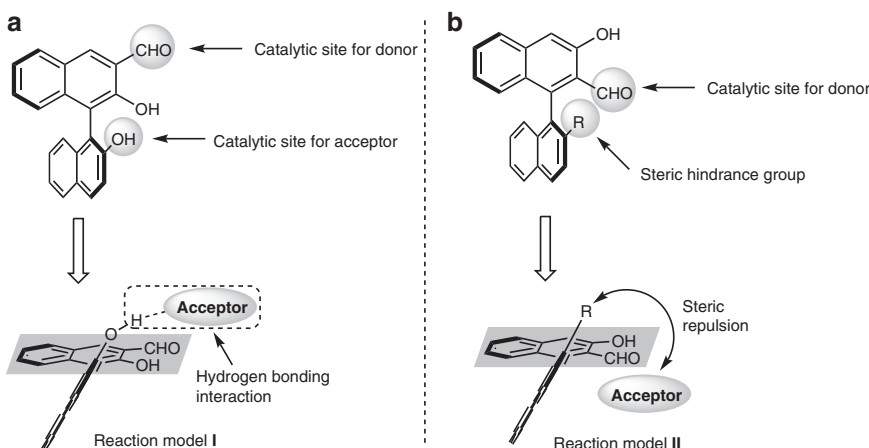

**Fig. 1 Different reaction models induced by chiral BINOL aldehyde catalysts. a** Reaction model **I** induced by 3-formyl BINOL aldehyde **I**. **b** Reaction model **II** induced by 2-formyl BINOL aldehyde **II**.

**Table 1 Reaction condition optimisation for 1,6-conjugated addition[a].**

| Entry | 3/4 | Time (h) | Yield (%)[b] | dr (syn:anti)[c] | ee (syn/anti) (%)[c] |
|---|---|---|---|---|---|
| 1 | 3a | 4 | 49 | 19:81 | -/33 |
| 2 | 4a | 3 | 36 | 64:36 | 58/- |
| 3 | 3b | 1.5 | 39 | 7:93 | -/72 |
| 4 | 3c | 2 | 54 | 13:87 | -/70 |
| 5 | 3d | 2 | 36 | 14:86 | -/77 |
| 6 | 3e | 3 | 36 | 15:85 | -/39 |
| 7 | 3f | 2 | 54 | 14:86 | -/45 |
| 8 | 3g | 3 | 44 | 18:82 | -/48 |
| 9 | 3h | 2 | 58 | 11:89 | -/44 |
| 10[d] | 3d | 5 | 64 | 10:90 | -/96 |
| 11[e] | 3d | 67 | 74 | 10:90 | -/97 |
| 12 | 4b | 2 | 33 | 80:20 | -31/- |
| 13 | 4c | 10 | 28 | 57:43 | 34/- |
| 14 | 4d | 3 | 37 | 62:38 | 60/- |
| 15 | 4e | 7 | 40 | 63:37 | 48/- |
| 16 | 4f | 6 | 48 | 73:27 | 50/- |
| 17 | 4g | 3 | 36 | 76:24 | 79/- |
| 18[f] | 4g | 5 | 37 | 79:21 | 83/- |
| 19[d,f] | 4g | 3 | 69 | 94:6 | 86/- |
| 20[f,g] | 4g | 7 | 70 | 95:5 | 90/- |
| 21[f,g,h] | 4g | 7 | 56 | 94:6 | 84/- |

[a]Reaction conditions: **1a** (0.1 mmol), **2a** (0.2 mmol), **3** or **4** (0.02 mmol), TMG (0.1 mmol), PhCH₃ (0.5 mL), at 25 °C.
[b]Isolated yield.
[c]Determined by chiral HPLC.
[d]Using ᵗBuOK as a base.
[e]Using 10 mol % **3d**, 30 mol % ᵗBuOK as base and 1 mL PhCH₃ as solvent.
[f]Using mesitylene as a solvent.
[g]Using 30 mol % ᵗBuOK as a base.
[h]Using 10 mol % **4g** as a catalyst.

conditions of the *anti*-selective addition were investigated first (Table 1, entries 3–11). Catalyst screening indicated that chiral aldehyde **3d** was optimal in terms of enantioselectivity (Table 1, entry 5). After we replaced the base TMG by potassium *tert*-butoxide (*t*BuOK), the yield and stereoselectivities of product *anti*-**5a** were enhanced greatly (Table 1, entry 10). When the catalyst loading was decreased to 10 mol %; comparable results were maintained (Table 1, entry 11).

Conditions of the *syn*-selective reaction were then optimised through screening catalysts, solvents and bases, as well as tuning the reaction temperature and reactant concentrations (see Supplementary Tables 2–4). The best results were obtained when chiral aldehyde **4g** was employed as a catalyst (Table 1, entry 17), mesitylene as solvent (Table 1, entry 18) and *t*BuOK as a base (Table 1, entry 19). Enantioselectivity was increased to 90% ee when the equivalency of *t*BuOK was decreased from 1 to 0.3 (Table 1, entry 20). When loading of chiral aldehyde **4g** was decreased to 10 mol %, product *syn*-**5a** was generated in 56% yield, 94:6 dr and 84% ee (Table 1, entry 21). According to these results, the reaction conditions depicted in entries 11 and 20 were selected as optimal and applied for substrate-scope investigation.

We first examined the scope of *para*-quinone methide and amino acid substrates for the *anti*-selective reaction. The scope of *para*-quinone methide derivatives that can be used as a substrate is quite broad (Fig. 2). All of the *para*-quinone methides bearing substituted phenyls, regardless of the degree of phenyl substitution, gave corresponding products in good-to-high yields (52–84%), high diastereoselectivities (88:12–>99:1) and enantioselectivities (94–98% ee) (Fig. 2, *anti*-**5b**–**5n**). No obvious effects on stereoselectivity from substituent position or electronics were observed. When the 2-(methoxycarbonyl)phenyl-substituted *para*-quinone methide was used as acceptor, an intramolecular amidation took place following the asymmetric conjugate addition, which gave product *anti*-**5l** in moderate yield and high diastereo- and enantioselectivity. Other aryl-substituted *para*-quinone methides, including naphthyl, thienyl and indolyl, were also investigated. Although enantioselectivity of *anti*-**5q** and diastereoselectivity of *anti*-**5r** were moderate, experimental outcomes were acceptable overall (Fig. 2, *anti*-**5o**–**5s**). Amino acids other than glycine ester could participate in this reaction and give high stereoselectivities, but higher catalyst loading (20 mol%) and long reaction time were needed. The yields were greatly affected by the alkyl substituents (Fig. 2, *anti*-**5t**–**5w**).

The substrate scope of the *syn*-selective reaction was then investigated. All the *para*-quinone methides investigated in the *anti*-selective reaction reacted smoothly with glycine ester **2a** in the presence of chiral aldehyde catalyst **4g**, giving the corresponding *syn*-**5b**–**5s** in good-to-high yields (50–83%), diastereoselectivities (70:30–98:2) and modest to high enantioselectivities (59–94% ee). However, no amino acids other than glycine esters reacted efficiently with *para*-quinone methide **1a** in this transformation, potentially as a result of the steric influence of the amino acids (Fig. 2, *syn*-**5t**–**5w**).

The absolute configurations of product *anti*-**5g** (*RR*, CCDC 1989580) was determined by X-ray single-crystal analysis, while that of *syn*-**5g** (*SR*) was assigned by comparing the specific rotation value with literature (see Supplementary Information). The stereoselective chemistries of other products **5** were assigned by analogy with those of *anti*-**5g** and *syn*-**5g** accordingly.

**Mannich reaction**. Use of the same diastereodivergent asymmetric catalytic strategy in multiple transformations provides a good indication of wider potential. To date, most of the reported diastereodivergent catalytic strategies are applicable to only one type of reaction. After successfully realising the diastereodivergent

catalytic asymmetric 1,6-conjugate addition, we sought to illustrate a second example of diastereodivergence using the same approach.

Transamination is an important chemical process in biological systems. It involves the conversion of an α-keto acid into an α-amino acid from by transaminase and co-enzyme pyridoxamine[42–44]. One of the most important intermediates in transamination is the ketimine that is formed from the α-keto acid and pyridoxamine; this ketimine can then convert into an aldimine via proton exchange[45,46]. The conversion between ketimine and aldimine illustrates how a carbonyl can activate the benzylic C–H bond of pyridoxamine via the formation of an imine and accelerate the subsequent deprotonation process. Inspired by this mode of reactivity, and deducing that pyridinylmethanamine can be viewed simply as the core unit of a pyridoxamine, we surmised that chiral aldehyde catalysis might be effective for the asymmetric α-functionalization of pyridinyl-methanamines. The chiral pyridinylmethanamine unit is found frequently in biologically active compounds[47–49], chiral ligands[50–53] and natural products[54–58], so the preparation of optically active pyridinylmethanamine derivatives is highly valuable work.

We chose pyridin-2-ylmethanamine **6a** and imine **7a** as model reactants and chiral aldehydes **3a** and **4a** as catalysts. With **3a**, *syn*-**8a** was generated with 89:11 dr (Table 2, entry 1). When catalysed by **4a**, no diastereoselectivity was observed in the formation of product **8a**, but the percentage of *anti*-isomer did increase to 48% (Table 2, entry 2). We then optimised the reaction conditions under the catalysis of chiral aldehydes **3** and **4**, respectively. Initial catalyst screening indicated that chiral aldehyde **3d** was the optimal catalyst for the *syn*-Mannich reaction in toluene (Table 2, entry 5), but subsequent solvent investigation showed that **3b** gave better experimental results than **3d** when dichloromethane (DCM) was used (Table 2, entries 13–14). Proceeding with catalyst **3b**, further reaction conditions such as the type of base, the equivalency of base and the reaction temperature were investigated (see Supplementary Tables 5–10). The product *syn*-**8a** could be obtained in 83% yield, 97:3 dr and 90% ee when carried out in DCM at 0 °C with 0.7 equivalents DBU as a base (Table 2, entry 16).

We then optimised reaction conditions using chiral aldehydes **4**. Catalyst screening showed that the steric effect of the R group had a great impact on both diastereoselectivity and enantioselectivity. Chiral aldehyde **4f**, bearing a 4-*tert*-butyl-phenyl substituent at its 2′ position, yielded product *anti*-**8a** in 75:25 dr and 88% ee (Table 2, entry 21). Other reaction conditions including type of base, equivalency of base, reaction temperature and reactant concentration were then screened (see Supplementary Tables 11–16). Good yield (71%) and stereoselectivity (16:84 dr., 93% ee) of *anti*-**8a** were obtained when the reaction was carried out in *o*-xylene (0.1 M concentration) at −10 °C, with 0.3 equivalents DBU as a base (Table 2, entry 25). Based on these results, the reaction conditions depicted in entries 16 and 25 were selected for the investigation of substrate scope.

In the *syn*-Mannich reaction, phenyl imines bearing single substituent on the phenyl ring displayed favourable reactivity, giving products *syn*-**8a**–**8g** in good yields (59–83%), diastereoselectivities (95:5–>99:1) and enantioselectivities (82–90% ee). No obvious effects on stereoselectivity from substituent position or electronics were observed. Aldimines having two substituents at the 3,4-positions of phenyl ring were also suitable reactants, although dimethyl substitution provided for superior yield and selectivity compared to dichloro substitution (Fig. 3, *syn*-**8h** vs *syn*-**8i**). Other aryl imines, including naphthyl, thienyl and furyl aldimines, were then tested. Corresponding products **8j**–**8m** were generated in good yields, high diastereoselectivities and good-to-

**Fig. 2 Substrate scope of the 1,6-conjugated addition reaction[a].** [a]Reaction conditions: **1** (0.1 mmol), **2** (0.2 mmol), **3d** (0.01 mmol) or **4g** (0.02 mmol), $^t$BuOK (0.03 mmol), PhCH$_3$ (1.0 mL) or mesitylene (0.5 mL), at 25 °C. [b]Isolated yield. [c]Determined by $^1$H NMR. [d]Ee of the major diastereoisomer determined by chiral HPLC. [e]Using 20 mol % **3d**. [f]At 35 °C.

high enantioselectivities. Next, substituted pyridinylmethanamines were employed as donors. With respect to substituted pyridine-2-ylmethanamines, no matter the substituent installed at the C2, C3 or C4 position of the pyridine ring, products were obtained in good-to-high yields, enantioselectivities and diastereoselectivities (Fig. 3, *syn*-**8n**–**8p**). Pyridine-3-ylmethanamine gave product *syn*-**8q** in good yield, moderate diastereoselectivity and good enantioselectivity, while quinolin-2-ylmethanamine

produced *syn*-**8r** in high yield, diastereoselectivity and moderate enantioselectivity.

Next, the substrate scope of the *anti*-selective Mannich reaction was investigated. All of the substituted imines and pyridinylmethanamines investigated in the *syn*-Mannich reaction displayed good reactivity in the *anti*-selective reaction. The yields of the *anti*-products were affected by substituent sterics and electronics. Product yield decreased greatly when an imine

**Table 2 Reaction condition optimisation of the Mannich reaction[a].**

3a: R = Me    3l: R = 1-naphthyl
3b: R = Br    3m: R = 2-naphthyl
3c: R = Cl    3n: R = 9-anthryl
3d: R = CN    3o: R = 9-phenanthryl
3i: R = I
3j: R = CF$_3$
3k: R = 3,5-(CF$_3$)$_2$C$_6$H$_3$

4a: R = Ph
4b: R = H
4c: R = 2-naphthyl
4d: R = 4-PhC$_6$H$_4$
4e: R = 4-TMSC$_6$H$_4$
4f: R = 4-$^t$BuC$_6$H$_4$
4g: R = 3,5-$^t$Bu)$_2$C$_6$H$_3$

| Entry | 3/4 | Time (h) | Yield (%)[b] | dr (syn:anti)[c] | ee (syn/anti) (%)[c] |
|---|---|---|---|---|---|
| 1 | 3a | 24 | 74 | 89:11 | 12/- |
| 2 | 4a | 12 | 70 | 52:48 | 23/70 |
| 3 | ent-3b | 12 | 95 | 93:7 | −66/- |
| 4 | 3c | 12 | 95 | 92:8 | 65/- |
| 5 | 3d | 12 | 92 | 94:6 | 74/- |
| 6 | 3i | 12.5 | 86 | 94:6 | 61/- |
| 7 | 3j | 9 | 86 | 96:4 | 59/- |
| 8 | 3k | 12 | 74 | 91:9 | 8/- |
| 9 | 3l | 12 | 52 | 89:11 | 16/- |
| 10 | 3m | 12.5 | 71 | 87:13 | 0/- |
| 11 | 3n | 12.5 | 69 | 91:9 | 24/- |
| 12 | 3o | 12 | 73 | 91:9 | 26/- |
| 13[d] | ent-3b | 12 | 87 | 95:5 | −76/- |
| 14[d] | 3d | 17 | 87 | 94:6 | 75/- |
| 15[d,e] | ent-3b | 12 | 85 | 94:6 | −81/- |
| 16[d,e,f] | ent-3b | 66 | 83 | 97:3 | −90/- |
| 17 | 4b | 12 | 90 | 78:22 | -/48 |
| 18 | 4c | 12 | 64 | 55:45 | -/59 |
| 19 | 4d | 12 | 64 | 48:52 | -/74 |
| 20 | 4e | 12 | 64 | 32:68 | -/85 |
| 21 | 4f | 12 | 64 | 25:75 | -/88 |
| 22 | 4g | 12 | 51 | 45:55 | -/30 |
| 23[g] | 4f | 12 | 72 | 21:79 | -/89 |
| 24[g,h] | 4f | 44 | 72 | 17:83 | -/93 |
| 25[g,h,i] | 4f | 44 | 71 | 16:84 | -/93 |

[a]Reaction conditions: 6a (0.1 mmol), 7a (0.13 mmol), 3 or 4 (0.01 mmol), DBU (0.03 mmol), PhCH$_3$ (0.5 mL), at 20 °C.
[b]Isolated yield.
[c]Determined by chiral HPLC.
[d]0.5 mL DCM as a solvent.
[e]Using 0.07 mmol DBU.
[f]At 0 °C.
[g]At −10 °C.
[h]Using 0.5 mL of o-xylene as solvent.
[i]Using 1 mL of o-xylene as solvent.

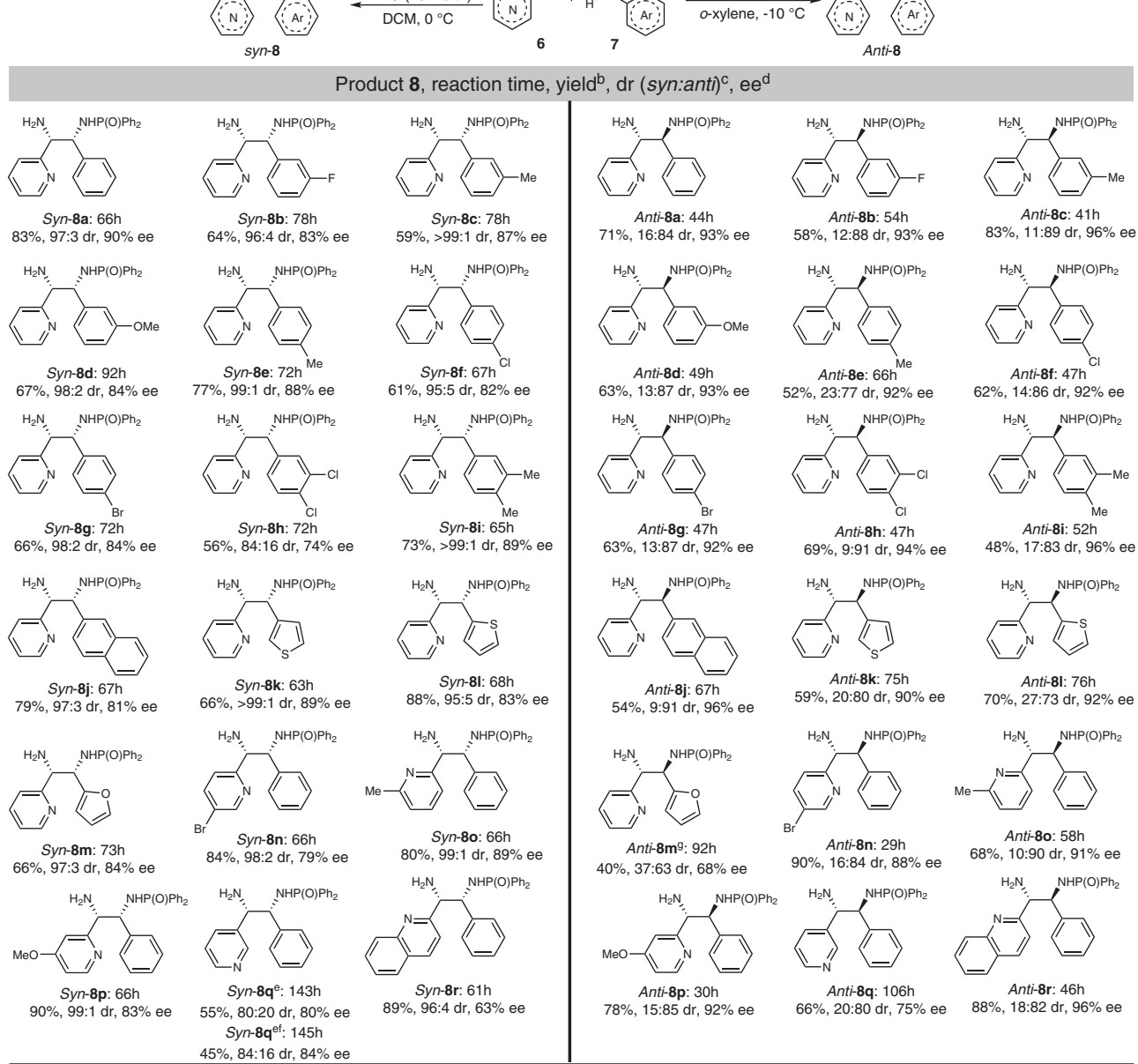

**Fig. 3 Substrate scope of the Mannich reaction[a].** [a]Reaction conditions: **6** (0.1 mmol), **7** (0.13 mmol), *ent*-**3b** or **4f** (0.01 mmol), DBU (0.07 or 0.03 mmol), $CH_2Cl_2$ (0.5 mL) or *o*-xylene (1.0 mL), at 0 °C or −10 °C. [b]Isolated yield. [c]Determined by [1]H NMR. [d]Ee of the major diastereoisomer determined by chiral HPLC. [e]Using toluene as solvent. [f]Using **3d** as a catalyst. [g]At 0 °C.

bearing an electron-rich aryl substituent was introduced as acceptor (Fig. 3, *anti*-**8i** and *anti*-**8m**). Overall, moderate-to-high diastereoselectivities (37:63–9:91) and enantioselectivities (68–96% ee) were observed in the *anti*-selective Mannich reaction; only three products had enantioselectivities lower than 90% ee.

The absolute configurations of products *anti*-**8a** (*RR*, CCDC 1989581) and *syn*-**8a** (*SR*, CCDC 1989582) were determined by the X-ray single-crystal analysis of their derivatives (see Supplementary Information). The stereoselective chemistries of other products **8** were assigned by analogy with those of *anti*-**8a** and *syn*-**8a** accordingly.

**Stereoselective control model investigation**. In order to clarify how the chiral aldehyde catalysts control stereoselectivity, we conducted comprehensive computational calculations to elucidate the possible reaction models of the 1,6-conjugate addition and Mannich reaction (see Supplementary Figs, 1–20 and Supplementary Data 1).

As our previous reported reaction process[30,31,33], the Schiff base intermediates formed by aminomethyl compounds and chiral aldehyde catalysts attacked *para*-quinone methide **1a** and aldimine **7a**, then the produced intermediates were hydrolysed to form **5a** and **8a**. Since the chiral centre was formed with the formation of the C–C bond, transition states (TSs) for the nucleophilic attack were individually investigated as a critical process for stereoselectivity. Because both reactions could be catalysed by **3b** and **4f**, the transition states for the reactions using them as catalysts were calculated.

After considering various conformations, the conformation with the lowest energy to form a different configuration of **5a** by

**Fig. 4 Calculated transition states of the conjugated addition reaction.** The structures of TSs to form **5a** by the 1,6-conjugate addition catalysed by **3b** and **4f** at the M06-2X/6-31(d) // M06-2X/6-31 + +G(d,p) level and energy are given in kcal/mol relative energy of **C-3b-RS-TS1** and **C-4f-SS-TS1**. Dr and ee values shown in the parentheses are the predicted data given by computational study.

the 1,6-conjugate addition catalysed by **3b** was calculated and shown in Fig. 4. We found the energy difference of **C-3b-RS-TS1** and **C-3b-SR-TS1** with the value of 1.41 kcal/mol. According to Van't Hoff equation, we predicted that the ee value was 83%, which is similar to the experimental value of 72%. As the energies of **C-3b-RR-TS1** and **C-3b-SS-TS1** were 1.52 kcal/mol and 2.31 kcal/mol higher than C-**3b-RS-TS1**, we predicted that the dr value was 92:8, which is compared with experimental data (93:7) in good agreement. By looking into the structures, there is an intermolecular hydrogen bond between the oxygen atom of **1a** and 2' hydroxyl of **3b** in **C-3b-RS-TS1**, which may play an important role in selectivity control. This inference was also supported by our experimental results: using the catalysts **3e**, **3f**, **3g** and **3h** with a big R group, that may affect the intermolecular hydrogen bond, resulting in worse ee values (see Table 1, entries 6–9). Additionally, these results indicate that the mechanism involving TS (transition state) **C-3b-RS-TS1** and leading to the final product with (RS)-configuration had the most favourable transition state structure.

On the other side, using **4f** as a catalyst led to the product with (SS)-configuration. Similarly, various conformations were calculated, and the conformation with the lowest energy for each configuration is shown in Fig. 4. It is likely that steric hindrance from 4-tert-butylphenyl would result in attack at Si face being more favourable. It is consistent with our experimental results: **4g** with a bigger steric hindrance group as 3,5-ditert-butylphenyl resulted in a better selectivity, while **4c** with a steric hindrance group as phenyl led to a wore one. In addition to steric effect, π–π interaction between the aryl ring of **1a** and binaphthyl also promotes the stereoselective formation of the SS configuration. Moreover, the predicted ee value based on the energy difference of **C-4f-SS-TS1** and **C-4f-RR-TS1** (0.59 kcal/mol) was 46% was slightly lower than the experimental result (50%). Also, the predicted dr value (77:23) is agreed with the experimental result (73:27).

Interestingly, the stereoselectivity for the Mannich reaction was different from the 1,6-conjugate addition when using the same catalyst **ent-3b** and **4f**. For Mannich reaction of **6a** and **7a** catalysed by **ent-3b**, the syn-**8a** was the main product, which anti-**5a** was the main product for the reaction of the 1,6-conjugate addition of **1a** and **2a**. To explain the stereoselectivity, the TS structures and energies for the additional step were calculated and were shown in Fig. 5. As the energy barrier of **M-ent-3b-RR-TS1** was 1.93 kcal/mol lower than **M-ent-3b-SS-TS1**, we predicted the ee value was 96%, which was slightly higher than the experimental results (90%). On the other hand, the predicted dr value (88:12) was lower than the experimental results (97:3). By looking into the structure, the intermolecular hydrogen bond between P=O group of **7a** and 2' hydroxyl of **ent-3b** with a distance of 1.62 Å may play an important role for the selectivity, because we also found that using catalysts **3k–3o** with a big R group at the 3' position that may affect the intermolecular hydrogen bond, resulted in worse ee values (Table 2, entries 8-12). In addition, reduction of steric repulsion and the π–π interaction between the C-phenyl of **7a** and the pyridinyl of **6a** in **M-ent-3b-RR-TS1** may also contribute to its lower energy. Compared to the TSs for 1,6-conjugate addition involving similar hydrogen bond (**C-3b-RS-TS1** and **M-ent-3b-RR-TS1**), this π–π interaction might be the might main difference and the possible reason for the different stereoselectivity.

Considering the nucleophilic addition of the Schiff base formed from catalyst **4f** and **6a** to aldimine **7a**, it is logical to think that steric hindrance from 4-tert-butylphenyl at the Re-face would again result in attack at Si face being more favourable. However, in this case, the experimental results showed that anti-**8a** (RS) was formed by Re-face attack. Therefore, we calculated various TS structures for both Re-face and Si-face attack and discovered that the energy barrier of **M-4f-RS-TS1** was the lowest. The energy of **M-4f-SR-TS1** was 1.64 kcal/mol higher than **M-4f-RS-TS1**. Based on it, the ee value for this reaction was predicted as 94%, which is

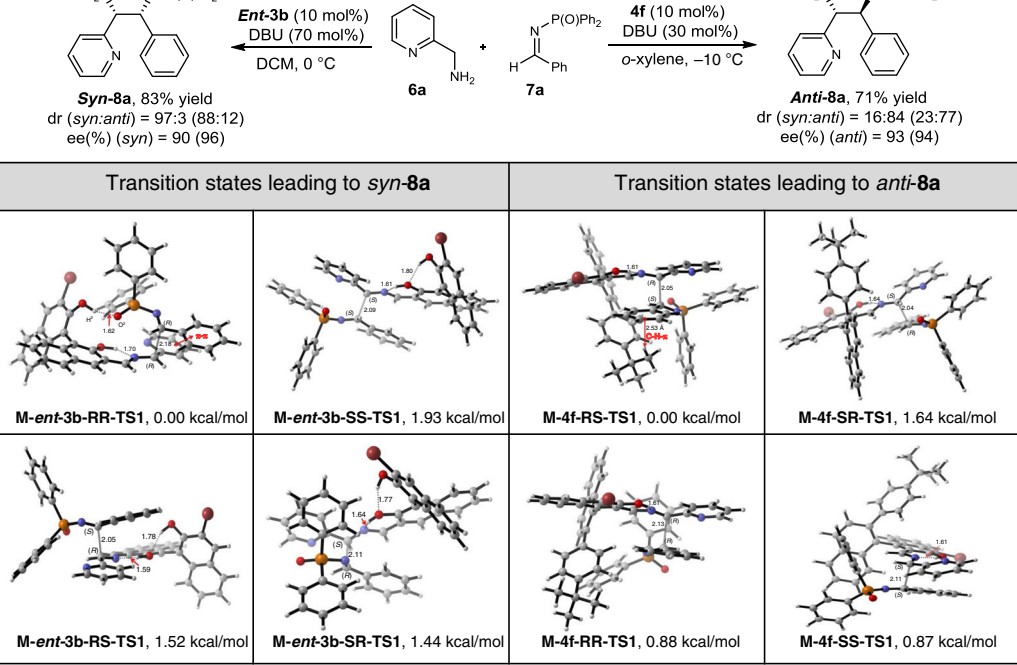

**Fig. 5 Calculated transition states of Mannich reaction.** The structures of TSs to form **8a** by Mannich reaction catalysed by **ent-3b** and **4f** at the M06-2X/ 6-31(d) // M06-2X/6-31 + +G(d,p) level and energy are given in kcal/mol relative energy of **M-ent-3b-RR-**TS1 and **M-4f-RS-TS1**. Dr and ee values shown in the parentheses are the predicted data by the computational study.

agreeing with the experimental results (93%). Additionally, the predicted dr value (77:23) was slightly lower than experimental results (84:16) as the energies of **M-4f-RR-TS1** and **M-4f-SS-TS1** was 0.88 kcal/mol and 0.87 kcal/mol higher than **M-4f-RS-TS1**, respectively. By looking to the structure, the angle between the *tert*-butyl and the conjugated pyridine plane was small and coincided with the C-phenyl ring of **7a**. Additionally, the distance between the H atom of *tert*-butyl group and C-phenyl ring of **7a** was just 2.53 Å, which suggests a potential favourable interaction between C–H and π in **M-4f-RS-TS1**. These results were also consistent with our catalyst screening results. Not only **4g** with a bigger steric hindrance group as 3,5-ditert-butylphenyl but also **4c** with a smaller group as phenyl, resulted into wore selectivities, suggesting the C–H and π interaction play an important role in the selectivity.

Based on these DFT calculation results, possible catalytic cycles for the four optimal model reactions were proposed. As shown in Fig. 6, all of these four reactions underwent via Schiff base formation, deprotonation, nucleophilic addition and hydrolysis process. In the 1,6-conjugated addition, chiral aldehyde **3** gave *syn*-selective products due to the formation of a hydrogen bond between the catalyst and *para*-quinone methide; while chiral aldehyde **4** was favoured to giving *anti*-selective products because of the steric effect and π–π interaction (Fig. 6a). With respect to the Mannich reaction, the hydrogen bond and π–π interaction formed between catalyst and imine induced the *syn*-selective products; while the steric effect and C–H–π interaction of catalyst **4** and imine promoted to form corresponding *anti*-Mannich products (Fig. 6b).

A 1,6-conjugate addition of amino acids to *para*-quinone methides and a Mannich reaction of pyridinylmethamines and aldimines were realised stereodivergently by chiral aldehyde catalysis. Chiral 3-formyl BINOL aldehyde catalysts were used to efficiently achieve the *anti*-1,6-conjugate addition and *syn*-Mannich reaction, while chiral 2-formyl BINOL aldehyde catalysts gave the *syn*-selective conjugate addition and *anti*-Mannich products. Generally, products of all transformations were obtained in good yield with high diastereo- and enantios-electivity. DFT calculations indicate that the orientation of formyl, hydroxyl and steric hindrance groups in chiral aldehydes **I** and **II** is the key factor in achieving diastereoselectively. It is promising for chemists to develop new stereodivergent reactions by chiral aldehyde catalysis under the guidance of these computational study results. The development of further stereo-divergent reactions by chiral aldehyde catalysis is ongoing in our lab.

## Methods

**Methods for the catalytic asymmetric 1,6-conjugated addition.** A dried 10 mL Schlenk tube was charged with *para*-quinone methides **1** (0.1 mmol), catalyst **3d** (0.01 mmol) or **4 g** (0.02 mmol), *t*BuOK (0.03 mmol) and α-amino acid derivatives **2** (0.2 mmol). Then dry solvent toluene (1.0 mL) or mesitylene (0.5 mL) was added to the tube, and the resulted mixture was effectively stirred at 25 °C for a suitable time. After the complete consumption of *para*-quinone methides **1** by TLC (Thin Layer Chromatograph), the solvent was removed by rotary evaporation. The crude mixture was subjected to [1]H NMR analysis to determine the diastereoselective ratio. Subsequently, a flash silica column-chromatography separation was performed for further purification. The details of the full experiments and compound characterisations are provided in the Supplementary Information.

**Methods for the catalytic asymmetric Mannich addition.** A dried Schlenk tube was charged with imine **7** (0.13 mmol), catalyst *R*-**3b** (0.01 mmol) or **4f** (0.01 mmol). Dry solvent CH₂Cl₂ (0.5 mL) or *o*-xylene (1 mL) was added, and the resulted solution was stirred at the indicated reaction temperature for 10 min. Then aminomethylpyridine **6** (0.1 mmol) and DBU (0.07 or 0.03 mmol) were added, and the resulted mixture was effectively stirred at the indicated temperature for a suitable time. After the complete consumption of aminomethylpyridines **6**, the solvent was removed by rotary evaporation. The crude mixture was subjected to [1]H NMR analysis to determine the diastereoselective ratio. Subsequently, a flash silica column-chromatography separation was performed for further purification. The details of the full experiments and compound characterisations are provided in the Supplementary Information.

**Fig. 6 Proposed catalytic cycles. a** The proposed catalytic cycles for the *anti*-selective 1,6-conjugated addition reaction (left) and *syn*-selective one (right). **b** The proposed catalytic cycles for the *syn*-selective Mannich reaction (left) and *anti*-selective one (right).

## Data availability

The X-ray crystallographic coordinates for structures reported in this study have been deposited at the Cambridge Crystallographic Data Centre (CCDC), under deposition numbers 1989580-1989582. These data can be obtained free of charge from The Cambridge Crystallographic Data Centre via www.ccdc.cam.ac.uk/data_request/cif. The authors declare that all other data supporting the findings of this study are available within the article and its Supplementary Information file.

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

## Acknowledgements

We are grateful for financial support from NSFC (22071199, 21472150), the Fundamental Research Funds for the Central Universities (XDJK2019AA003) and the Chongqing Science Technology Commission (cstccxljrc201701, cstc2018jcyjAX0548).

## Author contributions

W.W., L.M.J and G.Q.X. conceived this project. W.W and L.M.J carried out the experiments. Y.Y. and O.Q. performed the calculations. L.J.H. and W.Z.W. prepared chiral aldehyde catalysts and substrates. W.Z.L. and C.T. performed the HRMS analysis. O.Q. and G.Q.X. wrote the paper. All authors discussed the results.

## Competing interests

The authors declare no competing interests.
