## [Peer Review File · Nature Communications]

REVIEWER COMMENTS

Reviewer #1 (Remarks to the Author):

The manuscript by Prof. Guo and co-workers describes the development of a diastereodivergent strategy that utilizes chiral aldehyde catalysis. The study is a creative application of a pair of BINOL-derived aldehyde catalysts, developed in the author's research group, that generate complementary chiral spaces in asymmetric aldehyde catalysis. Pairs of catalysts are identified, after extensive optimization, that deliver diastereodivergent products for a 1,6-conjugate addition reaction and a Mannich reaction. Extensive substrate scope is reported for access to either diastereomer of product (in each reaction) in good yields and diastereo- and enantioselectivities. I expect this to be a high impact paper with a broad interest in the community; I recommend publication to Nature Communication after major revisions to the manuscript.

In the anti-selective 1,6-conjugate addition reaction of amino esters to para-quinone methides, 10 mol% catalyst loadings are used but in examples where an amino ester other than glycine ester is used, higher catalyst loadings (20 mol%) are required. There are some limitations to the syn-selective 1,6-conjugate addition reaction (Table 2) – specifically the relatively high catalyst loading (20 mol%) and the lack of reactivity of any amino-acid ester other than glycine esters. However, if one were to follow the same logic as the anti-selective reaction, the authors should attempt the 1,6-conjugate addition of amino-acid ester other than glycine esters using 40 mol% catalyst loading to see if the syn-selective reaction of these substrates can be achieved. Alternatively, the authors should try a less bulky catalyst such as 4f or 4e for the class of substrates that are unreactive in the syn-selective reaction.

Another nice addition to the paper would be a chemdraw scheme that outlines the proposed catalytic cycle for both reactions. This would better allow the reader to appreciate how aldehyde catalysis works for each of these reactions (rather than reading about the concept in text only). Apart from these minor comments, the reviewer finds that the experimental portion of this manuscript is well put together and illustrates some high-quality findings in this extremely important area of asymmetric catalysis.

On the other hand, the computational studies presented in the manuscript is much less impressive and lacks the scholarly narrative found in the experimental portion of the manuscript. Several aspects of the computational study are sub-par and according to the reviewer's opinion do not meet the standards of publication in Nature Communications. Some specific comments are:

- (1) Lacks a computational search of the lowest energy transition structures
- (2) Lack of correlation of calculated energies to experimental selectivities
- (3) Distances and key interactions not shown on the transition structure figures
- (4) No attempt has been made to make the reader understand or appreciate the origin of selectivity in each of the calculated systems

Points 3 and 4 can be addressed with major revisions to the manuscript. However, points 1 and 2 are of great concern and show lack of understanding of basic computational techniques when studying selectivities in these types of systems. A much more thorough study would have to be undertaken. "As is" the computational aspect of the manuscript does not add to the understanding of the experimental study, and actually detracts from the rigor of the experimental results.

1) Lacks a computational search of the lowest energy transition structures

A conformational search of the lowest energy transition structure in these systems is lacking. The author does not explain how they determined which structure should be used to represent each of the four possibilities (RR, RS, SR, SS). However, from the manuscript and supporting information the reviewer assumes that the authors did not conduct a thorough conformational search (only the structures shown in Fig 2 are listed in the SI). This is a fairly complex system that should have an

ensemble of possible transition structures leading to each intermediate. By not searching for the lowest energy transition structure, that leads to each of the four possible intermediates for each reaction, the error in predicting selectivities is great (leading to point #2).

2) Lack of correlation of calculated energies to experimental selectivities

The authors did a poor job of explaining exactly which set of calculations represented each experimental system. However, after looking carefully through the paper, the reviewer has found that the experimental data does not match-up well with the calculated energies. This fact is a bit buried (more on this in point #3). Each of the sets of calculations should show a clear correlation with energies and experimental selectivities. So, let's look at each system individually. For the set of calculations in figure 2a, this reviewer has assumed this represents Table 1 entry 3, the experimental dr is 7:93 and 72% ee. However, the calculated data is 4.0 kcal/mol between diastereomeric transition structures (a predicted dr of 0:100) and 4.86 kcal/mol between enantiomers (predicted 100% ee). The explanation for the large 4.0 kcal/mol difference between diastereomeric transition structures is "the orientation of the phenyl group may produce another pi-pi interaction between the aryl ring of 1a and the carbonyl group of 2a". This is a poor explanation of a 4.0 kcal/mol difference! The energy difference should be much smaller (1.5 kcal/mol for dr and 1.1 kcal/mol for ee), demonstrating that a thorough search of the phase space was probably not undertaken. (Also, the figures are too small to see any of the interactions discussed in the text - none of the interactions are highlighted or labeled in the figure).

Lack of correlation continues for the other systems as well. Fig 2b - (assuming Table 1 entry 16) though the dr is accurately predicted, the calculated energy for ee is 2.19 kcal/mol (95%ee) and the experimental ee is only 50%.

Fig 2c - (Table 3 entry 3) calculated energy for ee is 2.44 kcal/mol (97%ee) and the experimental ee is only 66%. Also of note, one of the diastereomers in fig 2c is 9.34 kcal/mol higher in energy. The reviewer finds it difficult to believe that a conformational search of the lowest energy transition structure leading to that intermediate would not lead to a lower energy transition structure.

Fig 2d - (Table 3 entry 21) predicted dr from calculations (44:56) experiment (25:75) and ee from calculations (100% ee) and 88% ee experimentally measured.

Though the reviewer acknowledges what calculations are not going to perfectly predict selectivities, there should be some consistencies and trends in the magnitudes. A study involving a thorough conformational search of the phase space should more correctly predict selectivities.

3) Distances and key interactions not shown on the transition structure figures and

4) No attempt has been made to make the reader understand or appreciate the origin of selectivity in each of the calculated systems

The theoretical section of the paper must be re-written for clarity. Authors should clearly connect the experimental data to the system being calculated. Figures are too small to see the structures or interactions clearly. The text on the figures is also too small and is difficult to read. The figures do not have any of the bond distances or interactions highlighted. I would suggest breaking figure 2 into five separate figures, increasing the size of the figures and adding proper labels. Figures and text from the calculational studies should describe the experimental data for comparison and clearly state which system is being studied (ie Table 1 entry 3).

Some important points that seem to be missing from the computational section -

How did you pick which systems to study computationally from the many experimental systems? Authors need to explain (using the figures and text) which interactions are favorable/disfavorable and contribute to selectivity in these systems.

What are the key results from this computational study that can be used for improving the reaction in the future?

Reviewer #2

This paper describes two new chiral aldehyde catalysed diastereodivergent reactions: 1,6-conjugate addition and Mannich reaction. As presented in the article, this is the first disclosure of diastereodivergence in chiral aldehyde catalysis. Both the syn- and anti-products could be obtained in moderate to high yields with excellent diastereo- and enantioselectivities. Additionally, this work provides DFT calculations to explain the proposed stereoselective control. Over all, this interesting reaction has the novelty enough to meet a broad readership from synthetic community. Therefore, I recommend publication in nature communication. However, some amendments would be necessary.

- 1) Considering the previous work (see reference 24), we want to know whether reducing the amount of catalyst can also give the same result.
- 2) Page 7, Table 2, there is no yield in syn-5 when $R1 \neq H$, we suggest adding $R1 = H$ to the formula.
- 3) Page 12, Table 4, are pyridine-2-ylmethanamines with electron-withdrawing groups and pyridine-4-ylmethanamines viable substrates in this reaction?

Response to Referees Letter

Dear referees,

Thanks for your valuable and expert comments for our Manuscript very much. Here we provided our response and revisions to your comments by point-to-point.

1. Reviewer 1's reports

Reviewer #1 (Remarks to the Author):

The manuscript by Prof. Guo and co-workers describes the development of a diastereodivergent strategy that utilizes chiral aldehyde catalysis. The study is a creative application of a pair of BINOL-derived aldehyde catalysts, developed in the author's research group, that generate complementary chiral spaces in asymmetric aldehyde catalysis. Pairs of catalysts are identified, after extensive optimization, that deliver diastereodivergent products for a 1,6-conjugate addition reaction and a Mannich reaction. Extensive substrate scope is reported for access to either diastereomer of product (in each reaction) in good yields and diastereo- and enantioselectivities. I expect this to be a high impact paper with a broad interest in the community; I recommend publication to Nature Communication after major revisions to the manuscript.

In the anti-selective 1,6-conjugate addition reaction of amino esters to para-quinone methides, 10 mol% catalyst loadings are used but in examples where an amino ester other than glycine ester is used, higher catalyst loadings (20 mol%) are required. There are some limitations to the syn-selective 1,6-conjugate addition reaction (Table 2) – specifically the relatively high catalyst loading (20 mol%) and the lack of reactivity of any amino-acid ester other than glycine esters. However, if one were to follow the same logic as the anti-selective reaction, the authors should attempt the 1,6-conjugate addition of amino-acid ester other than glycine esters using 40 mol% catalyst loading to see if the syn-selective reaction of these substrates can be achieved. Alternatively, the authors should try a less bulky catalyst such as 4f or 4e for the class of substrates that are unreactive in the syn-selective reaction.

Another nice addition to the paper would be a chemdraw scheme that outlines the proposed catalytic cycle for both reactions. This would better allow the reader to appreciate how aldehyde catalysis works for each of these reactions (rather than reading about the concept in text only).

Apart from these minor comments, the reviewer finds that the experimental portion of this manuscript is well put together and illustrates some high-quality findings in this extremely important area of asymmetric catalysis.

On the other hand, the computational studies presented in the manuscript is much less impressive and lacks the scholarly narrative found in the experimental portion of the manuscript. Several aspects of the computational study are sub-par and according to the reviewer's opinion do not meet the standards of publication in Nature Communications. Some specific comments are:

- (1) Lacks a computational search of the lowest energy transition structures
- (2) Lack of correlation of calculated energies to experimental selectivities
- (3) Distances and key interactions not shown on the transition structure figures
- (4) No attempt has been made to make the reader understand or appreciate the origin of selectivity in each of the calculated systems

Points 3 and 4 can be addressed with major revisions to the manuscript. However, points 1 and 2 are of great concern and show lack of understanding of basic computational techniques when studying selectivities in these types of systems. A much more thorough study would have to be undertaken. "As is" the computational aspect of the manuscript does not add to the understanding of the experimental study, and actually detracts from the rigor of the experimental results.

- 1) Lacks a computational search of the lowest energy transition structures

A conformational search of the lowest energy transition structure in these systems is lacking. The author does not explain how they determined which structure should be used to represent each of the four possibilities (RR, RS, SR, SS). However, from the manuscript and supporting information the reviewer assumes that the authors did not conduct a thorough conformational search (only the structures shown in Fig 2 are listed in the SI). This is a fairly complex system that should have an ensemble of possible transition structures leading to each intermediate. By not searching for the lowest energy transition structure, that leads to each of the four possible intermediates for each reaction, the error in predicting selectivities is great (leading to point #2).

- 2) Lack of correlation of calculated energies to experimental selectivities

The authors did a poor job of explaining exactly which set of calculations represented each experimental system. However, after looking carefully through the paper, the reviewer has found that the experimental data does not match-up well with the calculated energies. This fact is a bit

buried (more on this in point #3). Each of the sets of calculations should show a clear correlation with energies and experimental selectivities. So, let's look at each system individually. For the set of calculations in figure 2a, this reviewer has assumed this represents Table 1 entry 3, the experimental dr is 7:93 and 72% ee. However, the calculated data is 4.0 kcal/mol between diastereomeric transition structures (a predicted dr of 0:100) and 4.86 kcal/mol between enantiomers (predicted 100% ee). The explanation for the large 4.0 kcal/mol difference between diastereomeric transition structures is "the orientation of the phenyl group may produce another pi-pi interaction between the aryl ring of 1a and the carbonyl group of 2a". This is a poor explanation of a 4.0 kcal/mol difference! The energy difference should be much smaller (1.5 kcal/mol for dr and 1.1 kcal/mol for ee), demonstrating that a thorough search of the phase space was probably not undertaken. (Also, the figures are too small to see any of the interactions discussed in the text - none of the interactions are highlighted or labeled in the figure).

Lack of correlation continues for the other systems as well. Fig 2b – (assuming Table 1 entry 16) though the dr is accurately predicted, the calculated energy for ee is 2.19 kcal/mol (95%ee) and the experimental ee is only 50%.

Fig 2c – (Table 3 entry 3) calculated energy for ee is 2.44 kcal/mol (97%ee) and the experimental ee is only 66%. Also of note, one of the diastereomers in fig 2c is 9.34 kcal/mol higher in energy. The reviewer finds it difficult to believe that a conformational search of the lowest energy transition structure leading to that intermediate would not lead to a lower energy transition structure.

Fig 2d – (Table 3 entry 21) predicted dr from calculations (44:56) experiment (25:75) and ee from calculations (100% ee) and 88% ee experimentally measured.

Though the reviewer acknowledges what calculations are not going to perfectly predict selectivities, there should be some consistencies and trends in the magnitudes. A study involving a thorough conformational search of the phase space should more correctly predict selectivities.

3) Distances and key interactions not shown on the transition structure figures and

4) No attempt has been made to make the reader understand or appreciate the origin of selectivity in each of the calculated systems

The theoretical section of the paper must be re-written for clarity. Authors should clearly connect the experimental data to the system being calculated. Figures are too small to see the structures or

interactions clearly. The text on the figures is also too small and is difficult to read. The figures do not have any of the bond distances or interactions highlighted. I would suggest breaking figure 2 into five separate figures, increasing the size of the figures and adding proper labels. Figures and text from the calculational studies should describe the experimental data for comparison and clearly state which system is being studied (ie Table 1 entry 3).

Some important points that seem to be missing from the computational section –

How did you pick which systems to study computationally from the many experimental systems?

Authors need to explain (using the figures and text) which interactions are favorable/disfavored and contribute to selectivity in these systems.

What are the key results from this computational study that can be used for improving the reaction in the future?

2. Our response and revisions to Reviewer 1's comments:

Question 1: However, if one were to follow the same logic as the anti-selective reaction, the authors should attempt the 1,6-conjugate addition of amino-acid ester other than glycine esters using 40 mol% catalyst loading to see if the syn-selective reaction of these substrates can be achieved. Alternatively, the authors should try a less bulky catalyst such as 4f or 4e for the class of substrates that are unreactive in the syn-selective reaction.

Our response: Thanks for your useful suggestion. Four amino acid esters were selected as donors to react with para-Quinone methide by tuning the Ar substituent of catalyst 4 and the catalytic amounts of 4g. We found these reactions could not give the desired products with the reaction time of 4 days, so, the values of dr and ee were not determined (ND) (see Table R1).

Table R1: Experimental results given by catalysts 4.

syn-5	R ¹	R ²	4: Ar	x	yield (%)	dr (syn:anti)	ee (%)
5t	Me	Et	4a: C ₆ H ₄	20	trace	ND	ND
			4f: 4- ^t BuC ₆ H ₃	20	trace		

			4g: 3,5-2 ^t BuC ₆ H ₃	40	trace		
5u	Et	^t Bu	4a: C ₆ H ₄	20	0	ND	ND
			4f: 4- ^t BuC ₆ H ₃	20	0		
			4g: 3,5-2 ^t BuC ₆ H ₃	40	0		
5v	CH ₃ SCH ₂ CH ₂	Et	4a: C ₆ H ₄	20	0	ND	ND
			4f: 4- ^t BuC ₆ H ₃	20	0		
			4g: 3,5-2 ^t BuC ₆ H ₃	40	0		
5w	EtOOCCH ₂ CH ₂	Et	4a: C ₆ H ₄	20	trace	ND	ND
			4f: 4- ^t BuC ₆ H ₃	20	trace		
			4g: 3,5-2 ^t BuC ₆ H ₃	40	trace		

Question 2: Another nice addition to the paper would be a chemdraw scheme that outlines the proposed catalytic cycle for both reactions. This would better allow the reader to appreciate how aldehyde catalysis works for each of these reactions (rather than reading about the concept in text only).

Our response and revision: Thanks for your expert suggestion very much. We added corresponding catalytic cycles in our Revised Manuscript (see Figure 4 in the Revised Manuscript).

Question 3: Lacks a computational search of the lowest energy transition structures—A conformational search of the lowest energy transition structure in these systems is lacking. The author does not explain how they determined which structure should be used to represent each of the four possibilities (RR, RS, SR, SS). However, from the manuscript and supporting information the reviewer assumes that the authors did not conduct a thorough conformational search (only the structures shown in Fig 2 are listed in the SI). This is a fairly complex system that should have an ensemble of possible transition structures leading to each intermediate. By not searching for the lowest energy transition structure, that leads to each of the four possible intermediates for each reaction, the error in predicting selectivities is great (leading to point #2).

Our response and revisions: Thank for you expert suggestion about computational investigation. According to your suggestion, we did a thorough conformation search of transition structures to represent each of the four possibilities (RR, RS, SR, SS). The initial conformations of these

transition structures were generated by SYBYL-X2.0 using GA.conf. module, and then calculated by Gaussian 09. The transition structure employing the lowest energy, that leads to different stereoselective isomers, was calculated among 4-10 conformations. Four transition structures of each reaction possibilities were added into our Revised Supporting Information. The one with the lowest energy was revised in Manuscripts as figures 2-3. The methods for conformation search were added to SI.

Question 4: The authors did a poor job of explaining exactly which set of calculations represented each experimental system. However, after looking carefully through the paper, the reviewer has found that the experimental data does not match-up well with the calculated energies. This fact is a bit buried (more on this in point #3). Each of the sets of calculations should show a clear correlation with energies and experimental selectivities.

Our response and revisions: Thank for you expert suggestion to improve our work. We searched more conformations of TSs for each isomers. It is hard to say that all possible TSs had be calculated and found. However, we had used SYBYL-X2.0 using GA.conf. module to generate as more as possible initial poses and calculated the TSs based on these poses. So most conformations to form different isomers are different as we proposed before. The predicted dr value and ee for most reactions based on newly calculated energies are agreed with the experimental results.

Question 5: For the set of calculations in figure 2a, this reviewer has assumed this represents Table 1 entry 3, the experimental dr is 7:93 and 72% ee. However, the calculated data is 4.0 kcal/mol between diastereomeric transition structures (a predicted dr of 0:100) and 4.86 kcal/mol between enantiomers (predicted 100% ee). The explanation for the large 4.0 kcal/mol difference between diastereomeric transition structures is “the orientation of the phenyl group may produce another pi-pi interaction between the aryl ring of 1a and the carbonyl group of 2a”. This is a poor explanation of a 4.0 kcal/mol difference! The energy difference should be much smaller (1.5 kcal/mol for dr and 1.1 kcal/mol for ee), demonstrating that a thorough search of the phase space was probably not undertaken. (Also, the figures are too small to see any of the interactions discussed in the text - none of the interactions are highlighted or labeled in the figure.

Our response and revisions: For the reaction of 1a and 1b catalysed by 3b, after considering various conformations, the conformation with the lowest energy to form different configuration of 5a by the 1,6-conjugate addition catalysed by 3b was calculated and shown in figure 2 of the

Revised Manuscript. We found energy difference of C-3b-RS-TS1 and C-3b-SR-TS1 with value of 1.41 kcal/mol. According to Van 't Hoff equation, we predicted that the ee value was 83%, which is similar as the experimental value of 72%. As the energies of C-3b-RR-TS1 and C-3b-SS-TS1 were 1.52 kcal/mol and 2.31 kcal/mol higher than C-3b-RS-TS1, we predicted that the dr value was 11.3:1, which is compared with experimental data (93:7) in good agreement. By looking into the structures, there is an intermolecular hydrogen bonds between oxygen atom of 2a and 2' hydroxyl of 3b in C-3b-RR-TS1, which may play an important role in selectivity control. This inference was also supported by our experimental results: using the catalysts 3e, 3f, 3g, and 3h with a big R group, that may affect the intermolecular hydrogen bond, resulted in worse ee values. Additionally, these results indicate that the mechanism involving TS (transition state) C-3b-RS-TS1 and leading to the final product with (RS)-configuration had the most favourable transition state structure. These results and discussions can be found in the Figure 2 of our Revised Manuscript.

Question 6: Lack of correlation continues for the other systems as well. Fig 2b – (assuming Table 1 entry 16) though the dr is accurately predicted, the calculated energy for ee is 2.19 kcal/mol (95% ee) and the experimental ee is only 50%.

Our response and revisions: For the reaction of 1a and 1b catalysed by 4f, various conformations were calculated and the conformation with lowest energy for each configuration was shown in figure 3 of the Revised Manuscript. It is likely that steric hindrance from 4-tert-butyl-phenyl would result in attack at Si face being more favourable. It is consistent with our experimental results: 4g with a bigger steric hindrance group as 3,5-ditert-butylphenyl resulted in a better selectivity, while 4c with a steric hindrance group as phenyl led to a worse selectivity. In addition to steric hindrance, π - π interaction between the aryl ring of 1a and binaphthyl also promotes stereoselective formation of the SS configuration. Moreover, the predicted ee value based on the energy difference of C-4f-SS-TS1 and C-4f-RR-TS1 (0.59 kcal/mol) was 46% was slightly lower than the experimental result (50%). Also the predicted dr value (3.3:1) is agreed with experimental result (73:27). These results and discussions can be found in the Figure 2 of our Revised Manuscript.

Question 7: Fig 2c – (Table 3 entry 3) calculated energy for ee is 2.44 kcal/mol (97% ee) and the experimental ee is only 66%. Also of note, one of the diastereomers in fig 2c is 9.34 kcal/mol

higher in energy. The reviewer finds it difficult to believe that a conformational search of the lowest energy transition structure leading to that intermediate would not lead to a lower energy transition structure.

Our response and revisions: For the reaction of 6a and 7a catalysed by ent-3b using DCM as solvent under 0 °C, syn-8 was the main product with dr value of 97:3 and ee value of 90%. Because the energy barrier of M-ent-3b-RR-TS1 was 2.59 kcal/mol lower than M-ent-3b-SS-TS1. We predicted the ee value was 96%, which was slight higher than the experimental (90%). On the other hand, the predicted dr value (7:1) was lower than experimental results (97: 3). Although the predicted dr and ee value are not perfectly predict selectivities, there theoretical prediction and experimental results are consistencies in the magnitudes. These results and discussions can be found in the Figure 3 of our Revised Manuscript.

Question 8: Fig 2d – (Table 3 entry 21) predicted dr from calculations (44:56) experiment (25:75) and ee from calculations (100% ee) and 88% ee experimentally measured.

Our response and revisions: For the reaction of 6a and 7a catalysed by 4f using o-xylene as solvent under -10 °C, the energy of M-4f-SR-TS1 was 1.64 kcal/mol higher than M-4f-RS-TS1. Based on it, the ee value for this reaction was predicted as 94%, which is agreeing with the experimental results (93%). Additionally, the predicted dr value (77:23) was slightly lower than experimental results (84:16) as the energies of M-4f-RR-TS1 and M-4f-SS-TS1 was 0.88 kcal/mol and 0.87 kcal/mol higher than M-4f-RS-TS1, respectively. These results and discussions can be found in the Figure 2 of our Revised Manuscript.

Question 9: Though the reviewer acknowledges what calculations are not going to perfectly predict selectivities, there should be some consistencies and trends in the magnitudes. A study involving a thorough conformational search of the phase space should more correctly predict selectivities.

Our response: Thank for you expert suggestion to improve our work. After using computational search method and more TSs calculation, most predicted selectivities were agreed with the experimental results. These results and discussions can be found in the Figures 2-3 of our Revised Manuscript.

Question 10: Distances and key interactions not shown on the transition structure figures

Our response and revisions: Thanks for your suggestion very much. The key interactions which

affected the stereoselective results were shown in our Revised Manuscript (see Figures 2-3).

Question 11: No attempt has been made to make the reader understand or appreciate the origin of selectivity in each of the calculated systems. The theoretical section of the paper must be re-written for clarity. Authors should clearly connect the experimental data to the system being calculated. Figures are too small to see the structures or interactions clearly. The text on the figures is also too small and is difficult to read. The figures do not have any of the bond distances or interactions highlighted. I would suggest breaking figure 2 into five separate figures, increasing the size of the figures and adding proper labels. Figures and text from the calculational studies should describe the experimental data for comparison and clearly state which system is being studied (ie Table 1 entry 3).

Our response and revisions: Thank for you expert suggestion about computational investigation. We have revised figures 2-4 as your suggestion. The theoretical section was re-written and the figures were redone.

Question 12: Some important points that seem to be missing from the computational section – How did you pick which systems to study computationally from the many experimental systems? Authors need to explain (using the figures and text) which interactions are favorable/disfavorable and contribute to selectivity in these systems.

Our response and revisions: Thank for you expert suggestion about computational investigation. Chiral aldehydes ent-3b and 4f were the optimal catalysts for the Mannich reaction, so, the catalysts ent-3b and 4f involved optimal experimental systems of Mannich reaction were used as our computational investigation models. In order to give a more clearly comparison and draw a more reasonable conclusion, the experimental systems of 1,6-conjugated additions involved the same catalysts 3b and 4f were selected as computational study models. The key interactions leading to favorable/disfavorable transition states were explained in our Revised Manuscript.

Question 13: What are the key results from this computational study that can be used for improving the reaction in the future?

Our response: The results of our computational study clearly shown that the hydrogen bonding and steric hindrance were the two major factors that could affect the diastereoselectivity and enantioselectivity. By introducing suitable electrophiles with the site of hydrogen-bond donor, or by tuning the steric effect of chiral aldehyde catalyst, chemists can create more useful chiral

aldehyde-catalyzed stereodivergent reactions for the preparation of market valuable chiral amino acids and amines. We added this deducing in our Revised Manuscript.

3. Reviewer 2's reports

This reviewer is asking the following:

- 1) Considering the previous work (see reference 24), by reducing the amount of catalyst can also give the same result.
- 2) Page 7, Table 2, there is no yield in syn-5 when $R1 \neq H$, we suggest adding $R1 = H$ to the formula.
- 3) Page 12, Table 4, are pyridine-2-ylmethanamines with electron-withdrawing groups and pyridine-4-ylmethanamines viable substrates in this reaction?

4. Our response and revisions to Reviewer 2's comments:

Question 1: Considering the previous work (see reference 24), by reducing the amount of catalyst can also give the same result.

Our response: Thanks for your suggestion very much. We found the yield decreased greatly after reducing the catalyst loading. It is seemed that the catalytic activity of our chiral aldehyde is lower than that reported in ref. 24.

Question 2: Page 7, Table 2, there is no yield in syn-5 when $R1 \neq H$, we suggest adding $R1 = H$ to the formula.

Our response and revision: Thanks for your suggestion very much, we revised this issue as your suggestion.

Question 3: Page 12, Table 4, are pyridine-2-ylmethanamines with electron-withdrawing groups and pyridine-4-ylmethanamines viable substrates in this reaction?

Our response: Pyridine-2-ylmethanamine bearing a bromine at the 5-position of pyridine ring was used as donor in this reaction, producing product 8n in excellent yields and good stereoselectivities (see syn- and anti-8n of Table 4 in the Revised Manuscript). We found pyridine-4-ylmethanamine could participate in this reaction and give corresponding product in good yield, moderate to good diastereoselectivity and enantioselectivity (see the results listed below), but the products always contained some impurities and could not provide neat 1H NMR spectrums, so we did not report these results.

Besides the above revisions, we added 8 references in the Revised Manuscript. All of them have
 relativity to our work.

Hopefully, this revised version of our manuscript can meet the high requirement of Nature
 Communications.

Sincerely yours,

Best wishes,

Dr. Guo

REVIEWERS' COMMENTS

Reviewer #1 (Remarks to the Author):

The manuscript by Prof. Guo and co-workers describes the development of a diastereodivergent strategy that utilizes chiral aldehyde catalysis. The study is a creative application of a pair of BINOL-derived aldehyde catalysts, developed in the author's research group, that generate complementary chiral spaces in asymmetric aldehyde catalysis. I expect this to be a high impact paper with a broad interest in the community; I recommend publication to Nature Communication after minor revisions to the manuscript.

I am satisfied with the changes that Dr. Guo et al have made to the manuscript. Specifically, the conformational search in the computational portion of the study. I recommend a few minor revisions before publication.

The chemdraw in Figure 1 for the reaction models should be modified so that atoms are not overlapping. Specifically, the alcohol overlapping the aldehyde (maybe reduce the text font?).

In Figure 2 and 3 the bond distances are written directly on top of the molecule. Either put them in the white space or use arrows. In several places they are unreadable.

Response to Referees Letter

Dear referee,

Thanks for your valuable and expert comments for our Manuscript very much. Here we provided our revisions to your comments by point-to-point.

Part I. Reviewer's comments

The manuscript by Prof. Guo and co-workers describes the development of a diastereodivergent strategy that utilizes chiral aldehyde catalysis. The study is a creative application of a pair of BINOL-derived aldehyde catalysts, developed in the author's research group, that generate complementary chiral spaces in asymmetric aldehyde catalysis. I expect this to be a high impact paper with a broad interest in the community; I recommend publication to Nature Communication after minor revisions to the manuscript.

I am satisfied with the changes that Dr. Guo et al have made to the manuscript. Specifically, the conformational search in the computational portion of the study. I recommend a few minor revisions before publication.

The chemdraw in Figure 1 for the reaction models should be modified so that atoms are not overlapping. Specifically, the alcohol overlapping the aldehyde (maybe reduce the text font?).

In Figure 2 and 3 the bond distances are written directly on top of the molecule. Either put them in the white space or use arrows. In several places they are unreadable.

Part II. Our point-to-point revisions

Question 1: The chemdraw in Figure 1 for the reaction models should be modified so that atoms are not overlapping. Specifically, the alcohol overlapping the aldehyde (maybe reduce the text font?).

Our revision: This figure was re-edited. Now the hydroxyl and formyl groups are shown clearly.

Question 2: In Figure 2 and 3 the bond distances are written directly on top of the molecule.

Either put them in the white space or use arrows. In several places they are unreadable.

Our revisions: All of these Figures were re-edited. Now all of the data can be seen clearly.

Hopefully, this revised version of our manuscript can meet the high requirement of Nature Communications.

Sincerely yours,

Best wishes,

Dr. Guo